

# Representing the effects of stratosphere-troposphere exchange on 3D O₃ distributions in chemistry transport models using a potential vorticity based parameterization

Jia Xing[1, 2], Rohit Mathur[1], Jonathan Pleim[1], Christian Hogrefe[1], Jiandong Wang[1, 2], Chuen-Meei Gan[1],

Golam Sarwar[1], David C. Wong[1], Stuart McKeen[3]

[1]The U.S. Environmental Protection Agency, Research Triangle Park, NC 27711, USA
[2]School of Environment, Tsinghua University, Beijing, 100084, China
[3]Cooperative Institute for Research in Environmental Sciences, University of Colorado, Boulder, CO 809309, USA

*Correspondence to: Rohit Mathur (mathur.rohit@epa.gov)*

**Abstract.** Downward transport of ozone (O₃) from the stratosphere can be a significant contributor to tropospheric O₃ background levels. However, this process often is not well represented in current regional models. In this study, we develop a seasonally and spatially varying potential vorticity (PV)-based function to numerically assimilate upper tropospheric / lower stratospheric (UTLS) O₃ in a chemistry transport model. This dynamic O₃-PV function is parametrized based on 21-year ozonesonde records from World

Ozone and Ultraviolet Radiation Data Centre (WOUDC) with corresponding PV values from a 21-year Weather Research and Forecasting (WRF) simulation across the northern hemisphere from 1990 to 2010. The result suggests strong spatial and seasonal variations of O₃/PV ratios which exhibits large values in the upper layers and in high latitude regions, with highest values in spring and the lowest values in autumn over an annual cycle. The newly-developed O₃/PV function was then applied in the Community Multiscale Air Quality (CMAQ) model for an annual simulation of the year 2006. The simulated UTLS O₃ agrees much better

with observations in both magnitude and seasonality after the implementation of the new function. Considerable impacts on surface O₃ model performance were found in the comparison with observations from three observational networks, i.e., EMEP, CASTNET and WDCGG. With the new function, the negative bias in spring is reduced from -20 to -15% in the reference case to -9 to -1%, while the positive bias in autumn is increased from 1 to 15% in the reference case to 5 to 22%. Therefore, the downward transport of O₃ from upper layers has large impacts on surface concentration and needs to be properly represented in regional models.

**1. Introduction**

Accurate characterization of distribution of tropospheric ozone (O₃) is of great interest because it is not only impacts human ecosystem health, but also affects the earth-atmosphere radiation balance and global climate. In addition to its well-known sources from the photochemistry involving NOₓ and VOC, O₃ in the troposphere can also originate from stratosphere-to-troposphere exchange in tropopause folds and in cut-off lows (e.g., Danielsen,1968; Bamber et al., 1984; Holton et al., 1995) which become

important during periods of significant downward transport or deep stratospheric intrusions (Roelofs and Lelieveld, 1997; Stohl et al., 2000; McCaffery et al., 2004; Zanis et al., 2014; Neu et al., 2014). The contribution of stratosphere-to-troposphere exchange to troposphere column O₃ is estimated to be 20-50% of all sources across the globe (Lelieveld and Dentener, 2000) and the fraction is becoming larger due to reductions in O₃ precursor emissions and climate change (Collins et al., 2003). Quantification of O₃





contributions from the stratosphere-to-troposphere exchange is crucial for air quality management strategies.

Accurate $O_3$ vertical profiles are also essential for specifying lateral boundary conditions (LBCs) used in regional air quality simulations to capture the amount of $O_3$ imported to a region (Lam and Fu, 2010). A dynamical model representation of free-tropospheric $O_3$ variability becomes necessary for regional model applications conducted over large spatial domain and long

simulation periods wherein sufficient opportunities exist for exchange between the boundary layer and free troposphere. LBCs traditionally derived from global model simulations, often can suffer from uncertainties associated with the simulation biases at global scale and inconsistent downscaling methodologies. In order to reduce such uncertainties and to adequately represent variability in the free troposhpere, a dynamic representation of the upper tropospheric / lower stratospheric (UTLS) $O_3$ is desirable in regional models.

Potential vorticity (PV) has often been used as an indicator for air mass exchange between the stratosphere and the troposphere due to its strong positive correlation with $O_3$ and other tracers transported from the lower stratosphere to upper troposphere (Danielsen,1968; Browell et al., 1987; Ancellet et al., 1994). Previous studies used a linear scaling with a constant $O_3$/PV ratio to parameterize the influence of stratosphere-to-troposphere exchange on $O_3$ near the tropopause (Ebel et al, 1991; Carmichael et al, 1998; McCaffery et al, 2004). However, the reported $O_3$/PV ratios vary depending on location, season, and altitude. For example,

Browell et al (1987) suggested the average ratio between $O_3$ mixing ratios and PV is 50.2 ppb/PVu (1 PV unit = $10^{-6}$ $m^2$ K $kg^{-1}$ $s^{-1}$) from a study conducted over southern Nevada and California in April, 1984, while Ancellet et al (1994) suggested that the $O_3$ to PV ratios in the tropopause fold were 30-40 ppb/PVu at an European sounding station during November 1990. Carmichael et al (1998) suggested a $O_3$/PV ratio in the range 50-100 ppb/PVu at two sites in East Asia during spring, with smaller values at lower altitudes. Roelofs and Lelieveld (2000) reported a strong seasonal variation of $O_3$/PV ratio ranging from 36-76 ppb/PVu with the

largest in spring and the lowest in autumn based on MOZAIC data across North America and Europe. Clearly, the $O_3$/PV ratio varies significantly as a function of location, altitude and time. However, to date, the spatial and temporal variations of $O_3$/PV ratio have not been not well quantified.

This study aims to analyze the relationship between PV and UTLS $O_3$ over a large spatial and temporal scale and then to derive a spatially and temporally varying PV-based function to numerically assimilate UTLS $O_3$ in a chemistry transport model. The method

of parameterization of the $O_3$-PV function is described in section 2. The evaluation of the new function through a model simulation is presented in section 3, and summarized in section 4.

## 2. Method

### 2.1 Ozonesonde observations and PV data

A 21-year air quality simulation across the northern hemisphere from 1990 to 2010 has been conducted in our previous study (Xing

et al., 2015) by using the Weather Research and Forecasting (WRF) coupled with the Community Multiscale Air Quality (CMAQ) model. Throughout the simulation, WRF was nudged towards NCEP/NCAR Reanalysis data and NCEP ADP operational global surface and upper air observational weather data as described in Xing et al. (2015). The model domain covers the northern hemisphere (see Figure 1) with a horizontal grid of 108 km×108 km resolution and 44 vertical layers of variable thickness between the surface and 50 hPa (Mathur et al., 2012, 2014).

The model configuration and evaluation is detailed in Xing et al (2015). Anthropogenic emissions are derived from EDGAR (Emission Database for Global Atmospheric Research, version 4.2) and biogenic emissions are derived from GEIA (Global Emission Inventory Activity) (Guenther et al., 1995; Price et al., 1997). The model performance in the simulation of gaseous and





particle concentrations was evaluated thorough comparison with several ground observation networks.

Ozonesonde observations were obtained from the World Ozone and Ultraviolet Radiation Data Centre (WOUDC). A total of 44 sites (noted as red dots in Figure 1) across the northern hemisphere were selected after applying screening criteria which required at least one complete year covering all 12 months. The $O_3$/PV ratio used in this study is calculated by using the $O_3$ mixing ratio

from WOUDC divided by the PV estimated from WRF.

**2.2 Parameterization of $O_3$-PV correlations**

We parameterize the $O_3$/PV ratio as a combination of one spatial function and one temporal function, as shown below:

$$O_3/PV = F(spatial) \times G(temporal) \tag{1}$$

The spatial function is parameterized through an analysis of $O_3$/PV correlations with latitude and air pressure on an annual average

basis. The spatial distribution of the $O_3$/PV ratio at three vertical levels (58, 76 and 95 hPa) is presented in Figure S1. At the top layer (58hPa), most sites exhibit a consistent $O_3$/PV ratio around 80 ppb/PVu, while larger values are noted in low latitude (<20°N) regions where PV is small. At lower altitudes (i.e., 76, 95hPa), the $O_3$/PV ratio decreases to around 50 ppb/PVu. The $O_3$/PV ratio against latitude on each of the three levels is fitted by a 5$^{th}$ order polynomial function, as below:

$$F_{p_0}(lat) = a(0) + a(1) \times lat + a(2) \times lat^2 + a(3) \times lat^3 + a(4) \times lat^4 + a(5) \times lat^5; lat = abs(latitude)$$

$$\tag{2}$$

The values of $a(0)$- $a(5)$ are summarized in Table 1.

Decrease in the magnitude of the $O_3$/PV ratio with decreasing height is evident across the sites. In addition, significant decreases are exhibited in low latitude regions. Figure S2 displays the slope of the $O_3$/PV ratio vs. pressure at different latitudes. These slopes are estimated from the slope of the $O_3$/PV ratio at three vertical layers (i.e., 58, 76 and 95 hPa), i.e.

$$F(lat, p_x) = F_{p_0}(lat) + f(lat) \times (p_x - p_0); p_x \in (50hPa, 100hPa), p_0 = 58.56hPa \tag{3}$$

A 2$^{nd}$ order polynomial function is then used to fit the latitude-dependent decreasing rate of $O_3$/PV vs. pressure (i.e., vertical heights), as follows:

$$f(lat) = \min[(b(0) + b(1) \times lat + b(2) \times lat^2), 0] \tag{4}$$

The values of $b(0)$- $b(2)$ are summarized in Table 1.

To examine the pressure- and latitude- dependent spatial function, we compared the measurement-based and function-fitted $O_3$/PV values in Figure 2. Generally, the new function is able to capture the variation of $O_3$/PV ratio at different horizontal and vertical locations. Large values are seen in upper layers and in low latitude regions. The correlation coefficients (R) between measurement-based and function-fitted values are 0.5-0.9 and normalized mean biases (NMB) are within ±15%.

The temporal function is parameterized through an analysis of $O_3$/PV ratios using seasonal cycles derived from monthly mean

values. As seen in Figure 3, the $O_3$/PV ratio exhibits significant seasonal variations over an annual cycle. Consistent with Roelofs and Lelieveld (2000), the highest $O_3$/PV ratios occur in spring and the lowest ratios in autumn. A trigonometric function equation is used to represent the seasonality of the $O_3$/PV ratio, as below:

$$G(mfrc) = 1 + sin\left(12° \times (mfrc \times 30 + c(0))\right) \times c(1) \; ; mfrc = \frac{JulianDay}{365} \tag{5}$$

The values of $c(0)$ and $c(1)$ are summarized in Table 1. The fitted function generally captures the seasonal variation of $O_3$/PV, as

seen in Figure 3.

One should note that the parameterization is based on limited air pressure values, thus it is only applicable for conditions within the range from 50 to 100 hPa. By combining the spatial and temporal functions, the newly-developed $O_3$/PV function has been incorporated in the WRF-CMAQ model.





### 2.3 Application and evaluation

Simulations for the entire year of 2006 were conducted using the WRF-CMAQ model with the newly-developed $O_3/PV$ function (denoted as "Sim-new"), with a three month spin-up period (October-December in 2005). $O_3$ mixing ratios in the upper layers only (for air pressure smaller than 100hPa) are scaled based on the time varying modeled PV values and the function detailed in section

2.2. The simulated $O_3$ in the lower layers are then impacted by the modeled dynamics such as downwards transport.

A reference case (noted as "Sim-ref") is conducted by using a previously developed time- and space-invariant scaling ratio of 20 ppb/PVu derived from limited ozonesonde measurements from the IONS network and model PV estimates over the CONUS during summer 2006 (Mathur et al., 2008). The model simulations for which this 20 ppb/PVu scaling ratio was originally developed had a top level pressure of 100 hPa. Therefore, UTLS $O_3$ is expected to be underestimated in Sim-ref since the $O_3/PV$ should be larger

than 20 ppb/PVu when the top pressure of the model configuration is 50 hPa as in this study, particularly in high latitude regions and during spring time.

To investigate the influence of downward transport of $O_3$ through the stratosphere-to-troposphere exchange, we conducted a controlled case (noted as "Sim-off") in which no UTLS $O_3$ were simulated (i.e., the $O_3/PV$ function was turned off). The difference between Sim-new and Sim-off represents the impacts of downward transport of $O_3$ from the upper layers.

All other model options are identical in Sim-new, Sim-ref and Sim-off. The model configuration follows that described by Xing et al (2015). The anthropogenic emissions were derived from EDGAR. Biogenic VOC and lightning $NO_x$ emissions were obtained from GEIA. Sarwar et al. (2015) reported that halogen chemistry and enhanced $O_3$ deposition over sea-water can affect tropospheric $O_3$ mixing ratios. Thus, we included the impact of halogen chemistry and enhanced $O_3$ deposition in the marine environment in all three cases.

The observed data used for the evaluation is summarized in Table 2. The WOUDC global ozonesonde measurements in 2006 are used for comparison with the predicted $O_3$ vertical profile. Surface $O_3$ measurement networks, including the European Monitoring and Evaluation Programme (EMEP, http://www.emep.int), the Clean Air Status and Trends Network (CASTNET, http://epa.gov/castnet/) and the World Data Centre for Greenhouse Gases (WDCGG, http://ds.data.jma.go.jp/gmd/wdcgg/) including a few sites in Asia and Europe, are used for comparison with the predicted surface $O_3$ concentrations. Only data at sites

that covered more than 75% of entire year are selected for the comparison, except in the case of CASTNET because most sites have no $O_3$ records in winter (criteria set as at least 50% coverage).

### 3. Results and discussion

### 3.1 Comparison with O3 sonde data from WOUDC

Figure 4 displays the observed and simulated annual mean $O_3$ for three groups of layers, i.e., upper layers at pressure <100hPa,

middle layers at 300-500hPa and low layers at >800hPa, across the northern hemisphere. Seasonal comparisons for each group of layers are given in Figure S3-S5 (Spring = MAM, Summer = JJA, Autumn = SON, and Winter = DJF). Compared to the reference case (i.e., Sim-ref), the case with new-function (i.e., Sim-new) exhibits better agreement with observed UTLS $O_3$, especially in high latitude regions where $O_3$ levels are high. The statistics of model performance for simulated $O_3$ are summarized in Table 3. Both cases exhibit large R (>0.9) due to the strong correlation between UTLS $O_3$ and PV. However, the $O_3$ at high layers is

significantly underestimated in Sim-ref, especially during spring (by -75%) when the UTLS $O_3$ is the highest across the year. The application of the dynamic $O_3/PV$ function significantly improves the model performance for simulated $O_3$, as indicated by the significantly reduced NMB and the Normalized Mean Errors (NME) and slightly increased R-values in Sim-new. Such



improvements are also evident in the simulated $O_3$ in the middle layers. The NMB-range is reduced from about -37 to -17% in Sim-ref to about -17 to -1% in Sim-new. Changes in lower-layer $O_3$ are also noticeable. Because of the greater $O_3$ burden in upper layers, higher $O_3$ is seen at low layers in Sim-new compared to Sim-ref. Such increase of low-layer $O_3$ improves the model performance by reducing the negative biases in spring and summer from -15 and -9% in Sim-ref to -5 and -1% in Sim-new, but

deteriorates the model performance in Autumn as positive biases increase from 9% in Sim-ref to 15% in Sim-new. In winter, such increase of low-layer $O_3$ results in an overestimation of 13% in Sim-new from an underestimation of -7% in Sim-ref.

The observed and simulated $O_3$ vertical profiles at the 44 WOUDC ozonesonde monitor sites are compared in Figure 5. The simulated $O_3$ in Sim-new has better agreement with the WOUDC observations across most layers and latitudes. The most significant improvement is seen in the high latitude regions (latitude>45°) where the UTLS $O_3$ is significantly underestimated in

Sim-ref by up to -75% (Figure 5d), as the NMB is reduced to a moderate level (within ±30%) in Sim-new. The NME in Sim-new is reduced by 10% - 70% (Figure 5f), though the overestimation of near-surface $O_3$ in low latitude regions (latitude<45°) becomes even more evident in the Sim-new due to more downward transport of $O_3$ from the upper layers.

Comparisons of $O_3$ seasonal variations are displayed in Figure 6. With the application of dynamic $O_3$/PV function, the Sim-new results in better performance (with larger R values) in seasonality as well as magnitude at all layers for both low and high latitude

regions compared to the Sim-ref case. More importantly, the dynamic $O_3$/PV function derived from the seasonal variation of UTLS $O_3$, also improves the model performance in the simulation of the seasonality of low-layer $O_3$, suggesting the importance of accurate representation of downward transport of UTLS $O_3$ to the ground. In summer, the slight overestimation of high-layer $O_3$ but underestimation of middle-layer $O_3$ is noticeable in high latitude regions, which might be associated with the underestimation of downward transport of $O_3$ among those layers. A better vertically resolved model structure with higher resolutions may improve

representation of dynamics near the tropopause and help reduce such discrepancy in the simulation of downward transport of UTLS $O_3$.

### 3.2 Comparison with surface O3 observation networks

Increased UTLS $O_3$ burden leads to more downward transport of $O_3$ to the surface. To investigate such impacts on the model performance in simulation of ground-level $O_3$ concentrations, we compared the simulated surface $O_3$ from two cases, i.e., Sim-ref

and Sim-new, with observations from three surface networks. The observed surface daily maximum 8-h average $O_3$ (noted as "8h-max $O_3$"), the NMB in Sim-ref, and the difference between the NMEs from two simulations (i.e., NME in Sim-new minus Sim-ref) are presented in Figure 7 (for EMEP), Figure 8 (for CASTNET) and Figure S6 (for WDCGG). Statistics of the comparison are summarized in Table 3.

For EMEP sites in Europe, the 8h-max $O_3$ concentration in spring and summer is higher than in fall and winter. The Sim-ref

underestimates $O_3$ by -15% in spring but overestimates $O_3$ by +15% in autumn. The application of dynamic $O_3$/PV function reduces the low biases (NMB is reduced from -15% to -1%) in spring but increases the high biases (NMB is increased from +15% to +22%) in autumn, because of more $O_3$ downward transport from upper layers. In summer and winter, the overall performance in $O_3$ simulations changes from slight underestimation (NMB = -7% to -0.3%) to slight overestimation (NMB = +3% to +15%). In most cases, better performance in Sim-new is noticeable at sites where the $O_3$ is underestimated in Sim-ref (See Figure 7).

Similar results are presented at CASTNET sites across the United States, as seen in Figure 8. The 8h-max $O_3$ in the simulation of Sim-ref is largely underestimated by -17% and -10% respectively during spring and summer when the $O_3$ is the highest across the year. The sim-new exhibits relatively better results, as the NMB is reduced to -9% and -8% respectively in spring and summer (see Table 3). In addition, lower NMEs and larger Rs are noted in Sim-new compared to Sim-ref. During autumn, $O_3$ is mostly





overestimated at the eastern sites in the Sim-ref case and such overestimation becomes more pronounced in Sim-new, as the NMB is increased from +5% in Sim-ref to +8% in Sim-new. However, the performance for surface O₃ improved at the western U.S. sites in Sim-new. The O₃ in winter is underestimated by -10% in Sim-ref but overestimated by +6% in Sim-new.

At the WDCGG sites in Asia and Europe (see in Figure S6), the underestimation of O₃ during spring and winter is evident in Sim-ref. Better representation of UTLS O₃ and its subsequent downward transport in Sim-new improves the model performance, as the NMB is reduced from -20% (spring) and -17% (winter) to -8% (spring) and -0.03% (winter). The overestimation of O₃ in autumn becomes greater in Sim-new, as the NMB is increased from +0.8% to +5%.

### 3.3 Impacts on surface O3 from the downward transport

The relative change from the Sim-off to Sim-new, as displayed in Figure 9, can be used to investigate the influence of exchange process between the free troposphere and boundary layer on surface O₃ concentrations. High impacts are evident in high latitude regions where UTLS O₃ is high. Also the exchange between the free troposphere and boundary layer is much stronger during spring and winter. The increasing percentages from Sim-off to Sim-new in east China, the U.S., Europe and the northern hemisphere are estimated to be about 7.4%, 13.3%, 16.4% and 11.5%, respectively, on an annual basis. The downward transport from upper layers has even larger impacts on surface O₃ during spring and winter, and the increasing percentages in four regions are, respectively, about 10.5%, 17.0%, 21.0% and 15.1% in spring, and 15.4%, 25.5%, 32.3% and 20.4% in winter.

### 4. Summary and conclusion

A seasonally and spatially varying PV-based function was developed from an investigation of the relationship between PV simulated by WRF and UTLS O₃ derived from the WOUDC sonde observations over a 21-year period across the northern hemisphere. This new generalized dynamic O₃/PV function is successfully applied in the WRF-CMAQ model to numerically assimilate O₃ in the upper troposphere. The implementation of the new function significantly improves the model's performance in the simulation of UTLS O₃ in both magnitude and seasonality compared to observations, which then enables a more accurate simulation of the vertical distribution of O₃ across the northern hemisphere. These can then be used to derive more realistic vertically and temporally varying LBCs for regional nested model calculations.

Impacts on ground-level O₃ concentrations from the implementation of this dynamic function were evaluated through the comparison with three observation networks: EMEP, CASTNET and WDCGG. Compared to Sim-ref, more O₃ is transported downward to the surface due to a larger UTLS O₃ burden in Sim-new in which the dynamic O₃/PV function was applied. The implementation of the new dynamic function results in better performance in O₃ simulations in spring when O₃ is underestimated in Sim-ref relative to all networks, but worse performance in autumn when O₃ is overestimated in Sim-ref at most sites; the overestimation in autumn in Sim-ref suggests that processes other than free-troposphere-to-boundary-layer exchange are dictating this model bias.

The O₃ downward transport has strong spatial and temporal variations, as its impact is more evident in high latitude regions and during spring. Impacts of the downward transport can contribute to the background surface O₃ concentrations by 7.4%, 13.3%, and 16.4% in east China, the U.S. and Europe, respectively, on an annual basis and with even larger contributions in spring. Thus, the improvement of UTLS O₃ simulation is important to provide a better assessment of background O₃ levels.



**Acknowledgements**

The authors gratefully acknowledge the free availability and use of datasets from the WOUDC, WDCGG, CASTNET, EMEP monitoring networks. During the conduct of this work, JX and CG held National Research Council post-doctoral fellowships JW was a visiting student at the U.S. Environmental Protection Agency.

**Disclaimer:** Although this work has been reviewed and approved for publication by the U.S. Environmental Protection Agency, it does not necessarily reflect the views and policies of the agency.

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





**Table 1: Parameterization of the $O_3$/PV ratio correlation function**

| Nth of vector | | 0 | 1 | 2 | 3 | 4 | 5 |
|---|---|---|---|---|---|---|---|
| | p (hPa) | | | | | | |
| Horizontal (a) | 58.56 | 203.53 | -13.622 | 0.54157 | $-9.4264 \times 10^{-3}$ | $7.2990 \times 10^{-5}$ | $-2.0214 \times 10^{-7}$ |
| | 76.40 | 151.22 | 15.762 | 0.86918 | $1.8049 \times 10^{-2}$ | $1.8418 \times 10^{-4}$ | $-7.1408 \times 10^{-7}$ |
| | 95.62 | 62.217 | -1.4435 | $3.0439 \times 10^{-2}$ | $5.8000 \times 10^{-4}$ | $-1.5410 \times 10^{-5}$ | $8.2912 \times 10^{-8}$ |
| Vertical (b) | 58 to 96 | -2.1902 | $4.5507 \times 10^{-2}$ | $-2.4557 \times 10^{-4}$ | | | |
| Temporal (c) | 58 to 96 | 2 | 0.22 | | | | |



**Table 2: Summary of ozone observations used for comparison with WRF-CMAQ simulations**

|  | Network | Region | Number of sites | Record frequency | Date Sources |
|---|---|---|---|---|---|
| Vertical profile | WOUDC | Global | 44 | Several hourly data per month | http://www.woudc.org/ |
| Ground | WDCGG | Global | 12 | Hourly | http://ds.data.jma.go.jp/gmd/wdcgg/ |
|  | US-CASTNET | United States | 51 | Daily 8-hour maxima | http://epa.gov/castnet/ |
|  | EU-EMEP | Europe | 117 | Hourly | http://www.emep.int/ |





**Table 3: Statistics of model performances in ozone simulations**

| Network | Season | | | Sim-ref | | | | Sim-new | | | |
|---|---|---|---|---|---|---|---|---|---|---|---|
| Vertical hour-specific O$_3$ | | N pairs | Obs (ppb) | MB (ppb) | NMB (%) | NME (%) | R | MB (ppb) | NMB (%) | NME (%) | R |
| WOUDC- High layers (pressure<100hPa) | Spring | 249 | 1516.53 | -1135.39 | -74.87 | 67.43 | 0.90 | 26.07 | 1.72 | **21.05** | **0.94** |
| | Summer | 243 | 1176.71 | -776.68 | -66.01 | 58.97 | 0.93 | 84.47 | 7.18 | **22.58** | 0.93 |
| | Autumn | 246 | 1155.66 | -763.60 | -66.08 | 59.30 | 0.92 | **-30.03** | **-2.60** | **17.67** | **0.96** |
| | Winter | 243 | 1458.05 | -1073.49 | -73.63 | 63.86 | 0.91 | **-77.13** | **-5.29** | **23.25** | **0.95** |
| WOUDC- Middle layers (300hPa<pressure<500hPa) | Spring | 504 | 72.27 | -26.52 | -36.70 | 34.63 | 0.67 | **-9.71** | **-13.44** | **21.78** | *0.61* |
| | Summer | 492 | 73.08 | -19.79 | -27.09 | 27.93 | 0.56 | **-12.06** | **-16.51** | **21.37** | **0.70** |
| | Autumn | 498 | 59.45 | -9.99 | -16.80 | 18.91 | 0.66 | **-4.29** | **-7.21** | **16.97** | **0.68** |
| | Winter | 487 | 58.07 | -15.28 | -26.30 | 24.45 | 0.67 | **-0.60** | **-1.04** | **18.52** | **0.74** |
| WOUDC- Low layers (pressure>800hPa) | Spring | 1260 | 43.15 | -6.65 | -15.41 | 26.48 | 0.52 | **-0.43** | **-1.00** | **22.98** | 0.52 |
| | Summer | 1222 | 39.65 | -3.45 | -8.71 | 27.43 | 0.72 | **-2.10** | **-5.28** | 25.84 | **0.74** |
| | Autumn | 1241 | 34.66 | 3.22 | 9.28 | 25.91 | 0.61 | *5.18* | *14.93* | 28.89 | **0.62** |
| | Winter | 1229 | 33.80 | -2.23 | -6.60 | 56.00 | 0.33 | 4.33 | 12.80 | 63.92 | **0.40** |
| Surface 8h-max O$_3$ | | N pairs | Obs (μg m$^{-3}$) | MB (μg m$^{-3}$) | NMB (%) | NME (%) | R | MB (μg m$^{-3}$) | NMB (%) | NME (%) | R |
| EU-EMEP | Spring | 10539 | 95.22 | -14.31 | -15.03 | 20.71 | 0.46 | **-0.67** | **-0.70** | **17.73** | *0.41* |
| | Summer | 10663 | 96.44 | -0.24 | -0.25 | 22.29 | 0.66 | 2.62 | 2.71 | 22.36 | 0.66 |
| | Autumn | 10469 | 69.26 | 10.07 | 14.54 | 34.33 | 0.55 | *14.89* | *21.50* | *40.33* | *0.53* |
| | Winter | 10173 | 64.65 | -4.82 | -7.45 | 35.63 | 0.46 | 9.85 | 15.23 | 45.33 | **0.48** |
| US-CASTNET | Spring | 3919 | 113.03 | -19.54 | -17.28 | 20.76 | 0.52 | **-9.61** | **-8.50** | **15.69** | **0.53** |
| | Summer | 4456 | 115.58 | -11.36 | -9.83 | 19.48 | 0.62 | **-9.48** | **-8.20** | 18.81 | **0.63** |
| | Autumn | 3279 | 89.21 | 4.60 | 5.16 | 22.21 | 0.59 | *7.54* | *8.45* | *23.70* | *0.58* |
| | Winter | 1094 | 89.23 | -9.09 | -10.18 | 16.46 | 0.47 | 5.09 | 5.71 | **15.31** | 0.47 |
| WDCGG | Spring | 1023 | 114.50 | -22.46 | -19.61 | 23.52 | 0.53 | **-9.34** | **-8.16** | **17.25** | **0.57** |
| | Summer | 1021 | 100.30 | -1.62 | -1.62 | 33.93 | 0.55 | 0.80 | 0.79 | **32.57** | **0.60** |
| | Autumn | 948 | 91.22 | 0.72 | 0.79 | 29.38 | 0.46 | *4.55* | *4.99* | *30.42* | 0.46 |
| | Winter | 939 | 84.84 | -14.16 | -16.69 | 36.49 | 0.49 | 0.02 | 0.03 | **36.16** | *0.48* |

Note: **bold**= better performance in Sim-new than Sim-ref; *italic*= worse performance in Sim-new than Sim-ref



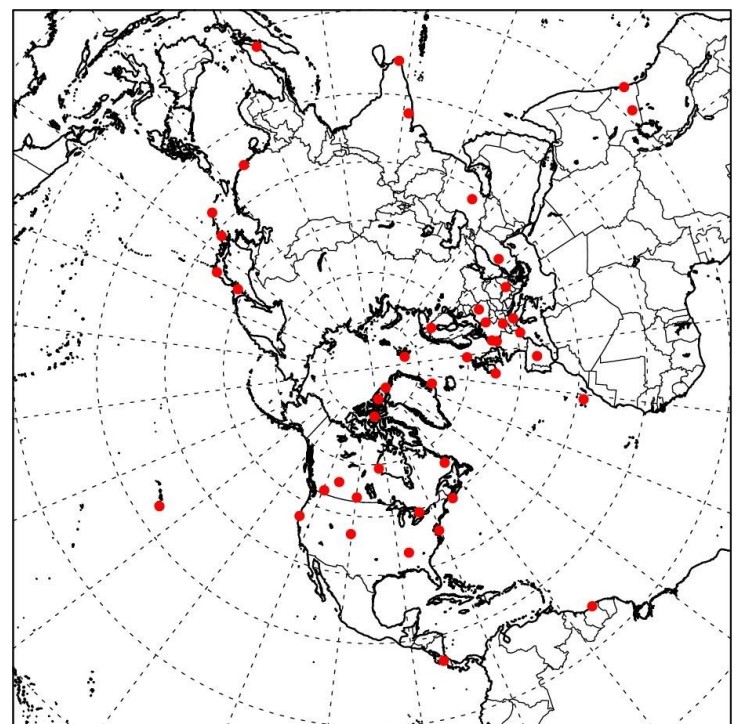

**Figure 1: Simulation domain and spatial locations of the 44 WOUDC sites used in this study**





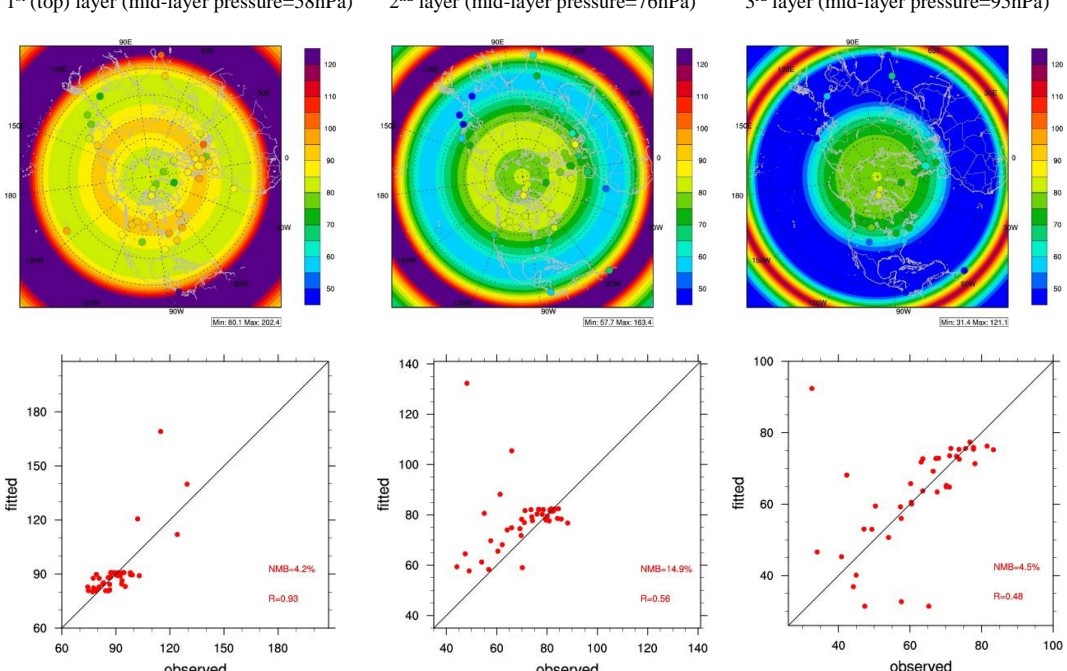

**Figure 2: Sensitivity of O₃/PV to spatial location (ppb/PVu)**



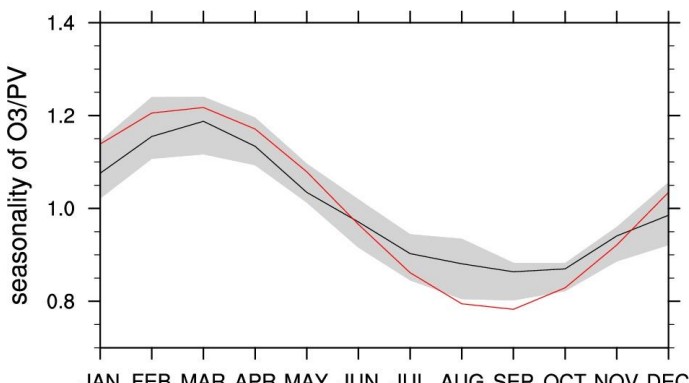

**Figure 3: Seasonality of O₃/PV (Annual mean= 1, black line=observed mean, red line = fitted by function, grey shadows represent 25% to 75% of observed records, selected pressure < 60 hPa)**





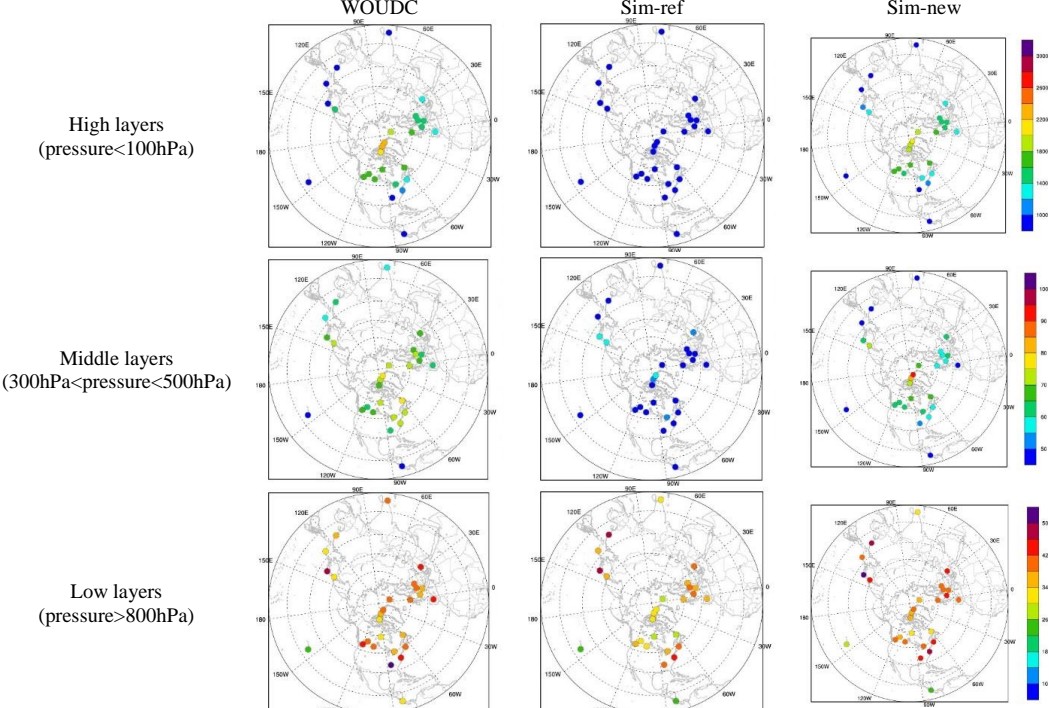

**Figure 4: Ozone spatial distributions (annual mean of measurement time period for each WOUDC site, ppb, 2006 Jan-Dec)**





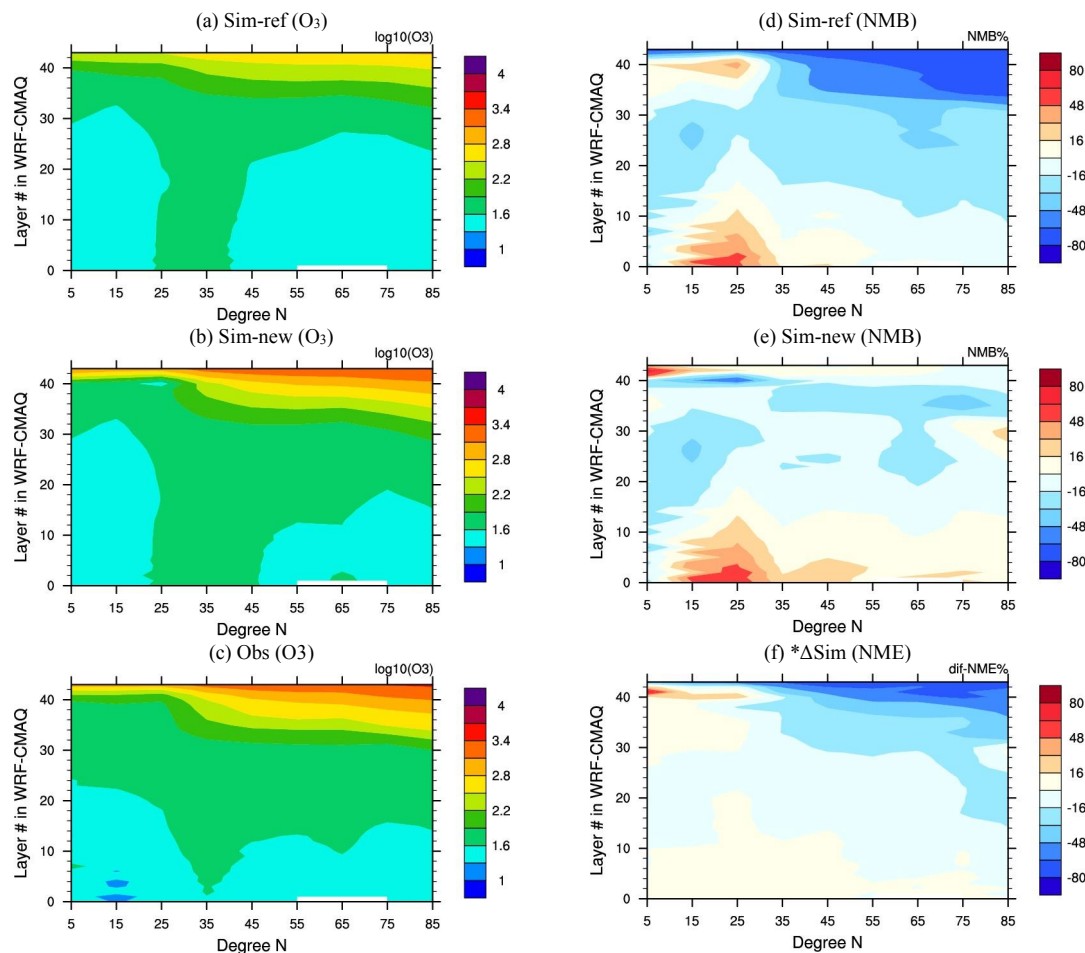

**Figure 5: Comparison of ozone vertical profiles in ozonesonde observations and WRF-CMAQ simulations (annual mean of measurement time period for each WOUDC site across the northern hemisphere, ppb, 2006 Jan-Dec; NMB- Normalized Mean Biases; NME-Normalized Mean Errors; *ΔSim= Sim-new minus Sim-ref in NME)**




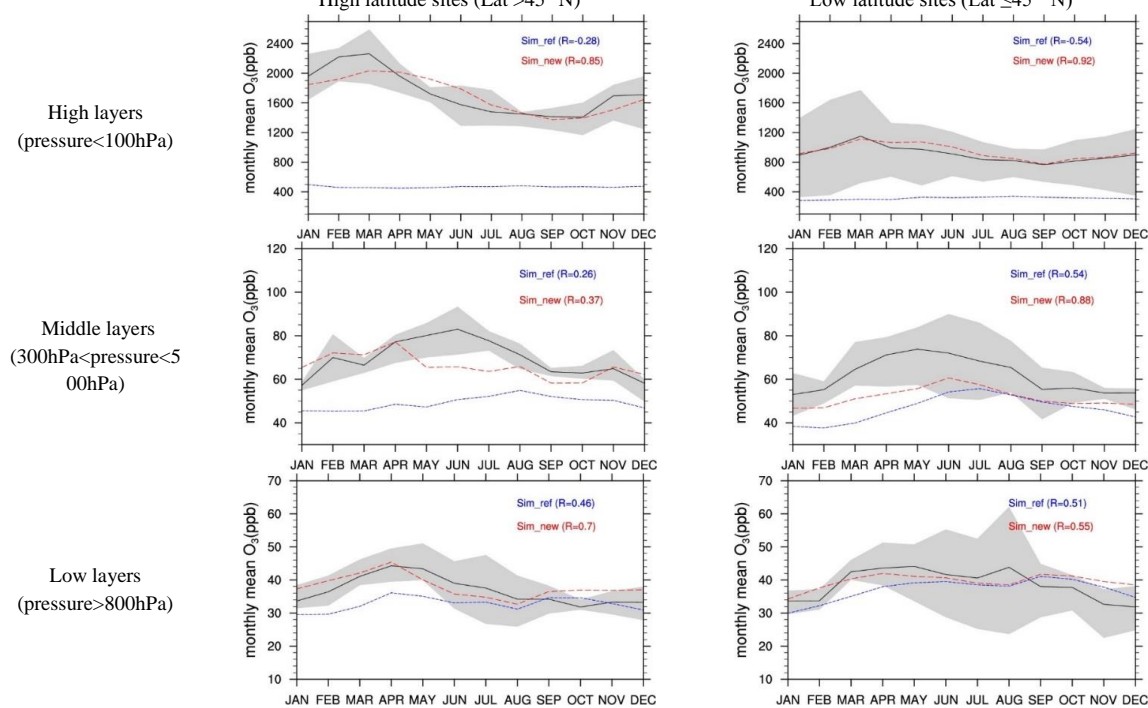

**Figure 6: Ozone seasonal variations (44-site averages of monthly mean of measurement time period for each WOUDC site, ppb, 2006 Jan-Dec, black lines represent observations, blue lines represent Sim-ref, red lines represent Sim-new, grey shadows represent 25th and 75th percentiles of observations)**





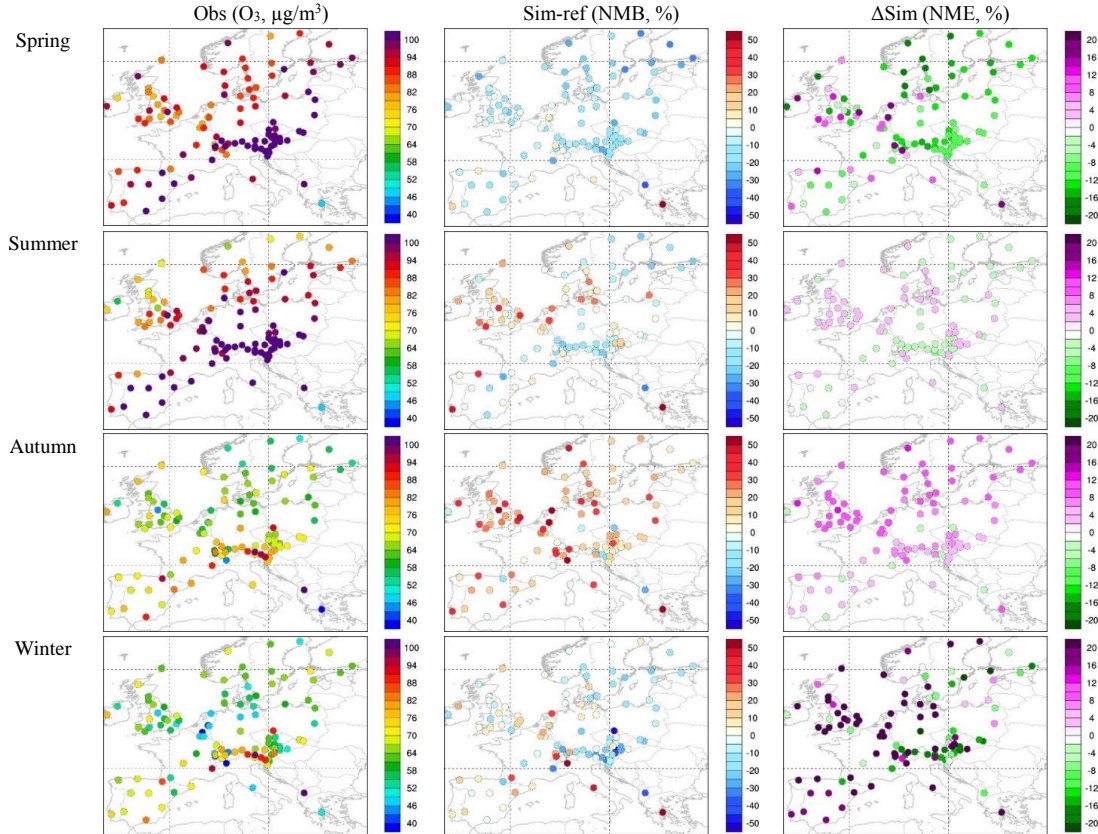

**Figure 7: Comparison with EMEP surface daily maximum 8-h average O₃ concentrations (ΔSim= Sim-new minus Sim-ref in NME)**





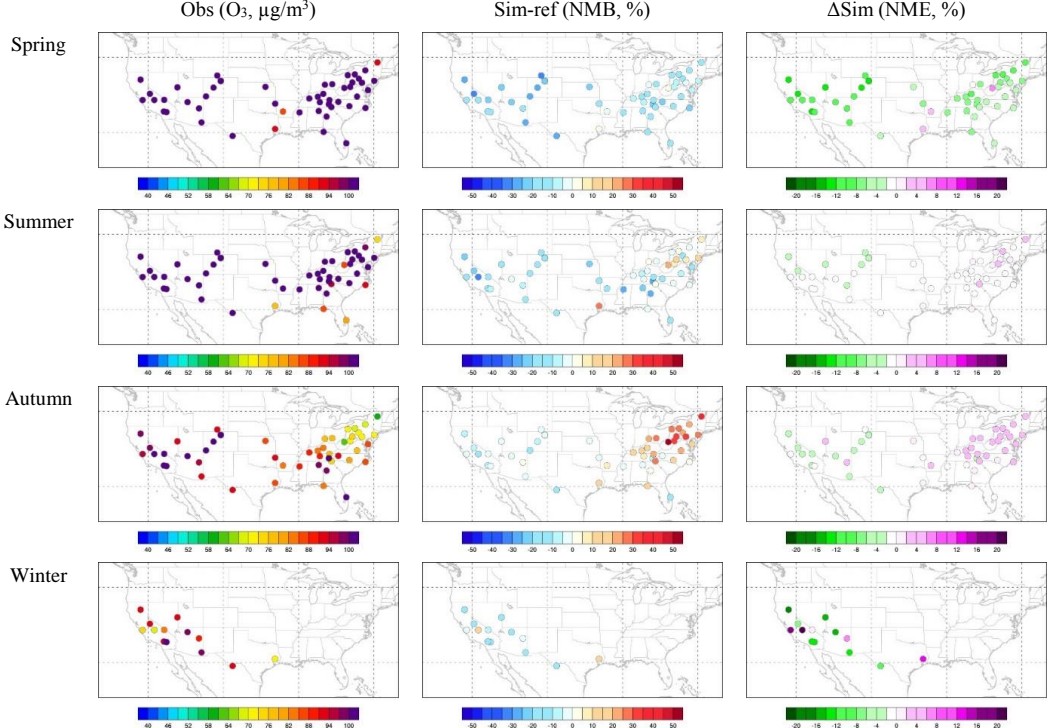

**Figure 8: Comparison with CASTNET surface daily maximum 8-h average O₃ concentrations (ΔSim= Sim-new minus Sim-ref in NME)**



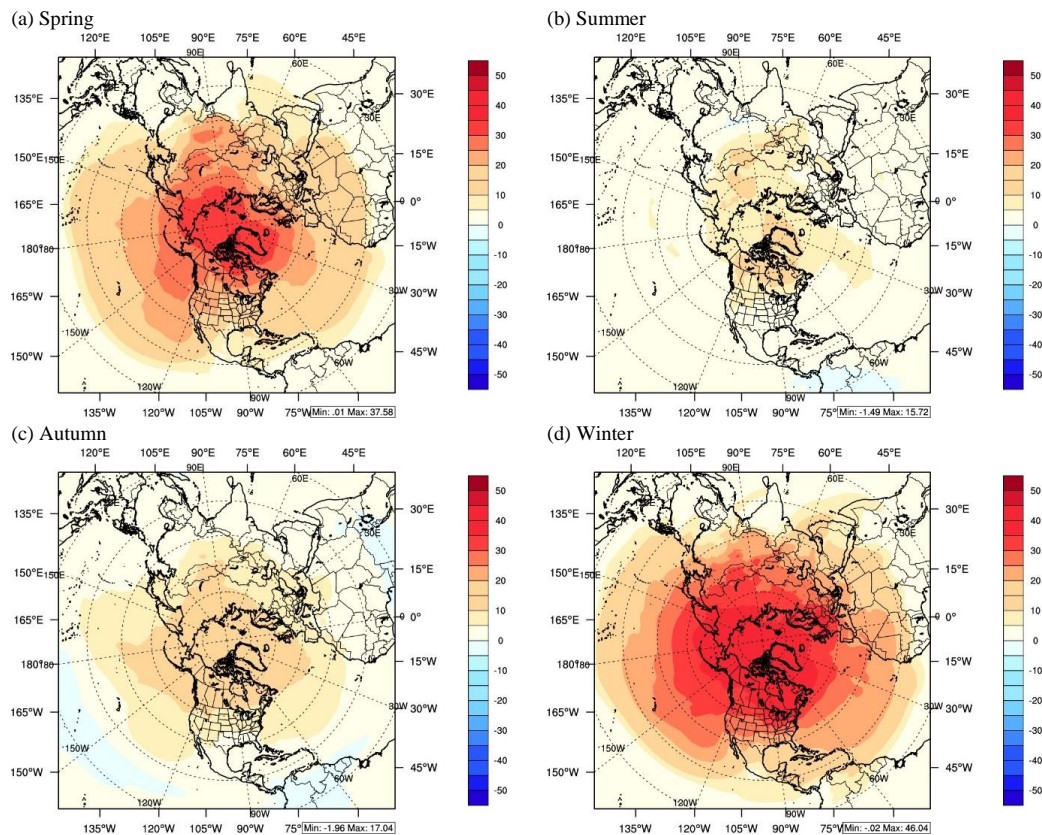

**Figure 9: Relative changes in surface O$_3$ due to dynamic O$_3$/PV function (unit:%, \*changes = (Sim-new minus Sim-off) / Sim-new )**