# Peer review of "Representing the effects of stratosphere-troposphere exchange on 3D O3 distributions in chemistry transport models using a potential vorticity based parameterization"

_Atmospheric Chemistry and Physics, 2016_

## Referee Comment (RC1) · Anonymous Referee #1 · 2 May 2016

General Comments:

The manuscript introduces a new parameterization for representing vertical, latitudinal, and seasonal variations in upper tropospheric/lower stratosphere (UT/LS) ozone within regional air quality modeling systems. The parameterization is based on regressions between modeled potential vorticity (PV) and observed ozone profiles. Observed ozone is based on measurements from 44 northern hemisphere World Ozone and Ultraviolet Radiation Data Centre (WOUDC) sites. The modeled PV is based on a 21 year (1990-2010) coupled Weather Research and Forecasting (WRF) Community Mul-

tiscale Air Quality (CMAQ) model. The new parameterization is able to account for the significant spatial and temporal variation in O3/PV ratios above 100 hPa and thus provides a much more generalized approach then used in previous studies. The impact of the new parameterization is evaluated by comparing a set of 1 year (2006) WRF-CMAQ simulations with WOUDC and surface measurements. Results show that the new parameterization significantly reduces low biases in the UT/LS compared to simulations with a fixed O3/PV ratio of 20 ppb/PVu resulting in positive impacts at the surface in spring. However, the new parameterization increased the high bias in surface ozone during autumn, resulting in negative impacts during this period. The methodology for developing the new parameterization, results, and conclusions are clearly presented and the work is highly relevant to the air quality modeling community. Figures and Tables in the main body of the manuscript are appropriate as are the supplemental figures.

Specific Comments:

The O3/PV parameterization relies on the assumption that both O3 and PV are conserved on planetary and synoptic transport time-scales, which is appropriate at middle and high latitudes of the UT/LS. However, in the tropics, sub-grid-scale convective transport largely determines the vertical distribution of ozone while differential diabatic heating due to convective latent heating/cooling introduces a source of UT/LS PV. As a result, the slope of O3/PV verses pressure shows a great deal of scatter for latitudes less than 30N (Figure S2 in the manuscript). This introduces significant uncertainties in the O3/PV regression in the tropical UT/LS and needs to be acknowledged. As a result, the new parameterization leads to increased Normalized Mean Errors (NME) compared to the reference simulation in the tropical UT/LS (Figure 5f in the manuscript). A discussion of the appropriateness of using the new O3/PV parameterization in the tropics needs to be included in the manuscript. Figure 5 (d) in the manuscript shows that the reference simulation Normalized Mean Bias (NMB) exhibits the classic "C" shaped signature of convective transport and suggests that overestimates in low-level ozone

lead to overestimates in tropical UT/LS ozone mixing ratios in the reference simulation. This should be discussed as well.

Technical Corrections:

Page 1 line 13: The PV based function does not result in assimilation of UT/LS O3 within WRF-CMAQ. I suggest changing "numerically assimilate" to "parameterize".

Page 1 line 14: Change "parameterized" to "developed"

Page 1 line 20: Change "new function" to "new parameterization"

Page 1 line 22: Change "new function" to "new parameterization"

Page 2 lines 10-26: Suggest adding a statement that co-variances between O3 and other species are not accounted for, which might introduce some inconsistencies in the chemistry

Page 2 line24: Change "numerically assimilate" to "parameterize"

Page 2 line 25: Change "parameterization" to "development"

Page 2 line 33: Add comment on how many vertical levels are above 100hPa

Page 3 line 35: Please comment on the overestimate in the amplitude of the seasonal cycle.

Page 3 lines 36-38: How well does the new parameterization handle LS ozone loss during Arctic springtime?

Page 5 lines 7-12: Please comment on the role of convective transport coupling the UT/LS and lower level overestimates (see specific comments)

Figure 4: Sim-new maps should be the same size as the WOUDC and Sim-ref maps.

[Figure]

---

## Referee Comment (RC2) · Anonymous Referee #2 · 18 May 2016

The authors derive an empirical correlation between stratospheric ozone and potential vorticity from two sets of data, with PV from the 21-year WRF simulation and ozone from WOUDC radiosondes during the same period. Specifically, they fit the ratio of ozone to PV as an order-5 polynomial function of latitude and an order-2 polynomial function of pressure (height). The temporal fitting, representing the seasonal variation, is done using a sine function with adjustable amplitude and phase. The spatial fitting is applied to the annually averaged data, so that the fitted model is given by a separable function of time and space. The applicable vertical domain is from 50-100 hPa. The parameterization is then applied to the WRF model to obtain the stratospheric ozone

concentration that couples to the air quality model (CMAQ) for year-2006 simulation.

The authors give two reasons why they did this study: (1) there is no stratospheric chemistry scheme available in the regional model (2) there is a wide range of measured O3/PV ratio values. The putative success of this parameterization is shown in the one-year simulation (Sim-new) where the results are compared to a reference simulation(Sim-ref) where O3/PV is simply set to be 20ppb/PVU.

My overall impression is that the authors didn't give much thought in formulating their parameterization (as I explain below). Nor did the authors try to articulate clearly why fitting polynomial functions of higher orders will be better than using the linear correlation in the context of what we already know about stratospheric dynamics and mixing, the photochemical source/sink of ozone and the processes that lead to non-conservation of PV. In my view, if the authors reviewed the stratospheric mixing literature, they could have designed better parameterization and experiments.

If the authors want to make the case that this is a simple paper based on "big data" statistical approach where the data are trained to relate one variable (PV) to another (ozone) by fitting polynomial functions with no need to discuss the underlying physics and atmospheric dynamics, the authors are then compelled to work out the uncertainty range of the coefficients in Table 1. Is the 20-year data long enough to obtain stable coefficients? If the data are split into two 10-year periods, are the coefficients very different from each other and if implemented in the simulation, would the results be very different from those in the 20-year run?

I suspect because the parameterization is confined to the region between 50 and 100 hPa, for Sim-new, the free tropospheric ozone concentrations are likely biased high, particularly worse in the winter when the tropopause is the lowest. Here is my reasoning. The tropopause height changes seasonally and STE occurs at the tropopause in connection with the lowermost stratosphere. The tropopause over winter mid- and high-latitudes can be as low as 300 hPa. In other words, the parameterization completely missed the lowermost stratosphere (LMS, defined as the volume enclosed by the 380K isentropic surface/or alternatively the 100hPa isobaric surface at the top and he tropopause below) where the isentropic mixing delivers air and ozone mass across the tropopause. In my view, if one feels compelled to parameterize ozone using PV, then the crucial region where this needs to be done well is the LMS region to capture the isentropic mixing. Even we assume that the Brewer-Dobson circulation transports the correct amount of ozone to the LMS when using the authors' parameterization, and that the model does the correct mixing, one would still expect that the resulting ozone would still be biased high because of the missing midlatitude photochemical ozone loss in the LMS.

For the reference case (Sim-ref) the specification of 20 ppb/PVU guarantees a large underestimation of stratospheric ozone as well as a large underestimation of ozone fluxes into the troposphere. It is very common in the literature that the tropopause is defined as the surface of PV= 2 PVU or Ozone =100 or 150 ppb of ozone. These isopleths are in close proximity. This is equivalent to 50-75 ppb/PVU near 150 to 300 hPa. Thus this sim-ref specification of 20 ppb/PVU is not only way too low for 50hPa as the authors already mentioned but is just too low in general. Another way to confirm my suspicion is to check the Sim-new's coefficients at zero-order (constant term) in Table 1, which I presume is the leading term. Indeed, they are 62, 151 and 203 ppb/PVU at 95, 76 and 58 hPa, much larger than 20 ppb/PVU, even near 100hPa.

Thus the major finding that Sim-new corrects the negative bias of Sim-ref but overcorrects it for the autumn and winter seasons is largely expected (Page 9).

It would be also helpful to see more comparisons of vertical profiles in the free troposphere alone between Sim-ref, Sim-new and WOUDC for the mean and variability of ozone. The log10(ozone) in Fig. 5 only gives a rough sense of magnitude dismatch, mainly that Sim-ref is too weak. The authors should zoom in on the free troposphere for more assessment. It is also better to present pressure height in km or pressure in hPa instead of model layers number in Fig. 5. And more efforts are needed for caption

descriptions.

Ultimately this paper is about surface ozone. Judged from Table 3, the surface ozone errors seem quite insensitive to the input of stratospheric ozone (Sim-ref vs Sim-new). Have the authors looked into the coupling scheme between the free troposphere and the planetary boundary layer in CMAQ? Any other option for representing the boundary-layer turbulence?

---

## Author Comment (AC1) · 20 Jul 2016

We thank the reviewer for the detailed and thoughtful review of our manuscript. Incorporation of the reviewer's suggestion has led to an improved manuscript. Detailed below is our response to the issues raised by the reviewer. We also detail the specific changes incorporated in the revised manuscript in response to the reviewer's comments.

[Comment]: The manuscript introduces a new parameterization for representing vertical, latitudinal, and seasonal variations in upper tropospheric/lower stratosphere

(UT/LS) ozone within regional air quality modeling systems. The parameterization is based on regressions between modeled potential vorticity (PV) and observed ozone profiles. Observed ozone is based on measurements from 44 northern hemisphere World Ozone and Ultraviolet Radiation Data Centre (WOUDC) sites. The modeled PV is based on a 21 year (1990-2010) coupled Weather Research and Forecasting (WRF) Community Multi-scale Air Quality (CMAQ) model. The new parameterization is able to account for the significant spatial and temporal variation in O3/PV ratios above 100 hPa and thus provides a much more generalized approach then used in previous studies. The impact of the new parameterization is evaluated by comparing a set of 1 year (2006) WRF-CMAQ simulations with WOUDC and surface measurements. Results show that the new parameterization significantly reduces low biases in the UT/LS compared to simulations with a fixed O3/PV ratio of 20 ppb/PVu resulting in positive impacts at the surface in spring. However, the new parameterization increased the high bias in surface ozone during autumn, resulting in negative impacts during this period. The methodology for developing the new parameterization, results, and conclusions are clearly presented and the work is highly relevant to the air quality modeling community. Figures and Tables in the main body of the manuscript are appropriate as are the supplemental figures.

[Response]: We thank the reviewer for the overall positive assessment of the manuscript and recognition of the implications of the results of the analysis presented.

[Comment]: The O3/PV parameterization relies on the assumption that both O3 and PV are conserved on planetary and synoptic transport time-scales, which is appropriate at middle and high latitudes of the UT/LS. However, in the tropics, sub-grid-scale convective transport largely determines the vertical distribution of ozone while differential diabatic heating due to convective latent heating/cooling introduces a source of UT/LS PV. As a result, the slope of O3/PV verses pressure shows a great deal of scatter for latitudes less than 30N (Figure S2 in the manuscript). This introduces significant uncertainties in the O3/PV regression in the tropical UT/LS and needs to be

acknowledged. As a result, the new parameterization leads to increased Normalized Mean Errors (NME) compared to the reference simulation in the tropical UT/LS (Figure 5f in the manuscript). A discussion of the appropriateness of using the new O3/PV parameterization in the tropics needs to be included in the manuscript.

[Response]: We agree with the reviewer that in the tropics, convective transport plays a significant role in shaping the vertical ozone profile which cannot be fully represented by the UTLS PV. The uncertainties in the O3-PV regression in the tropical UTLS as pointed by the reviewer is also suggested by the poor correlation between O3 and PV at latitudes south of 30N (Figure S2), resulting in an increased NME in Sim-ref in the tropical UTLS (Figure 5f). We agree with the reviewer that this aspect should be further elaborated in the discussions. Following the reviewer's suggestion, in revised manuscript, we provided additional discussion of the appropriateness of using the new O3/PV parameterization in the tropics, as below:

(Page 3 Line 31-33) "Poor correlation between O3 and PV is found for latitudes south of 30N (Figure S2). This is in part because in the tropics, convective transport plays a significant role in shaping the vertical ozone profile. Consequently, PV alone may not be able to robustly represent UT/LS O3 in the tropics."

(Page 5 Line 26-27) "An increased NME in Sim-ref is found in the tropical UT/LS, indicating the uncertainty in applying the new O3/PV parameterization in the tropics."

[Comment]: Figure 5 (d) in the manuscript shows that the reference simulation Normalized Mean Bias (NMB) exhibits the classic "C" shaped signature of convective transport and suggests that overestimates in low-level ozone lead to overestimates in tropical UT/LS ozone mixing ratios in the reference simulation. This should be discussed as well.

[Response]: We agree that the overestimates in both low-level and UT/LS ozone mixing ratios indicates the influence of the convective transport in the tropics. At the reviewer's suggestion, we provided additional discussion in the revised manuscript, as below:

(Page 5 Line 21-24) "The overestimates in both low-level and UTLS ozone mixing ratios exhibited in the C-shaped signature in the NMB in Sim-ref is also indicative of the likely influence of convective transport on three-dimensional O3 distributions in the tropics (e.g., Doherty et al., 2005), that is not adequately captured by the current parameterization."

Reference: Doherty, R. M., Stevenson, D. S., Collins, W. J., and Sanderson, M. G.: Influence of convective transport on tropospheric ozone and its precursors in a chemistry-climate model, Atmos. Chem. Phys., 5, 3205-3218, doi:10.5194/acp-5-3205-2005, 2005.

[Comment]: Page 1 line 13: The PV based function does not result in assimilation of UT/LS O3 within WRF-CMAQ. I suggest changing "numerically assimilate" to "parameterize".

[Response]: The "numerically assimilate" has been changed to "parameterize" in the revised manuscript.

[Comment]: Page 1 line 14: Change "parameterized" to "developed".

[Response]: The "parameterized" has been changed to "developed" in the revised manuscript.

[Comment]: Page 1 line 20: Change "new function" to "new parameterization"

[Response]: The "new function" has been changed to "new parameterization" in the revised manuscript.

[Comment]: Page 1 line 22: Change "new function" to "new parameterization"

[Response]: The "new function" has been changed to "new parameterization" in the revised manuscript.

[Comment]: Page 2 lines 10-26: Suggest adding a statement that co-variances between O3 and other species are not accounted for, which might introduce some inconsistencies in the chemistry

[Response]: The statement as suggested by the reviewer has been added in the revised manuscript as below:

(Page 2 Line 25-27) "One thing should be noted that such PV based parameterization only modifies the O3 mixing ratio, however, co-variances between O3 and other species are not accounted for in such modifications which might introduce some inconsistencies in the chemistry in the model's UTLS."

[Comment]: Page 2 line24: Change "numerically assimilate" to "parameterize"

[Response]: The "numerically assimilate" has been changed to "parameterize" in the revised manuscript.

[Comment]: Page 2 line 25: Change "parameterization" to "development"

[Response]: The "parameterization" has been changed to "development" in the revised manuscript.

[Comment]: Page 2 line 33: Add comment on how many vertical levels are above 100hPa

[Response]: The information of vertical levels is added in the revised manuscript as below:

(Page 2 Line 36-37) "44 vertical layers of variable thickness between the surface and 50 hPa (approximately 3 vertical levels above 100hPa)"

[Comment]: Page 3 line 35: Please comment on the overestimate in the amplitude of the seasonal cycle.

[Response]: The discrepancy in the amplitude of the seasonal cycle between the parametrization and the observations arises due to the difference in the number of sites these curves are representative of. The observed curve was based on data from
all sites, while the parameterized curve (red) utilized information only from locations north of 40N. We chose to base the seasonal variations only at sites >40N because that is where the seasonal variability was the strongest. We however agree that this is likely to cause some confusion. Thus we have revised figure 3 (see Figure C1) to now also include observed seasonality based on both (i) all sites, and (ii) sites at latitudes north of 40N.

We clarify it in the revised manuscript as below:

(Page 4 Line 2-6) "The seasonal variability in the O3-PV correlation also varies with latitude. The influence of convective transport on PV in the tropics, as discussed before, also results in weaker seasonal variations in the O3-PV correlation at low latitudes. This is seen in Figure 3, which compares the seasonal variations in this relationship inferred from (i) all sites, and (ii) sites at latitudes north of 40N. Thus to ensure that the parameterization more faithfully captures the seasonality at the higher latitudes, where it is strongest, we parameterize the temporal variations only on data at locations with latitudes north of 40N."

[Comment]: Page 3 lines 36-38: How well does the new parameterization handle LS ozone loss during Arctic springtime?

[Response]: Basically, the new parameterization is based on long-term observations, thus is able to capture the LS ozone loss which is reflected in the observation. We examined the arctic ozone trend over the past two decades and generally the new parameterization displays good performance in capturing the ozone level and its seasonal variability, as shown in Figure C2. However, it might not be able to capture the potential trend driven by factors other than PV, such as chemistry.

We have clarified this issue in the revised manuscript, as below:

(Page 4 Line 8-11) "Additionally, since only the O3-PV correlation is considered in the parameterization, the potential trend driven by factors other than PV (e.g., chemistry)

cannot be captured by the current parameterization. However, the effects of processes which are already reflected in the observation (e.g., seasonal variations such as lower stratosphere ozone loss during Spring) are implicitly captured in the parameterization."

[Comment]: Page 5 lines 7-12: Please comment on the role of convective transport coupling the UT/LS and lower level overestimates (see specific comments)

[Response]: The discussion of the role of convective transport to explain the overestimates of UT/LS and lower level O3 has been added in the revised manuscript, as below:

(Page 5 Line 21-24) "The overestimates in both low-level and UTLS ozone mixing ratios exhibited in the C-shaped signature in the NMB in Sim-ref is also indicative of the likely influence of convective transport on three-dimensional O3 distributions in the tropics (e.g., Doherty et al., 2005), that is not adequately captured by the current parameterization."

Reference:

Doherty, R. M., Stevenson, D. S., Collins, W. J., and Sanderson, M. G.: Influence of convective transport on tropospheric ozone and its precursors in a chemistry-climate model, Atmos. Chem. Phys., 5, 3205-3218, doi:10.5194/acp-5-3205-2005, 2005.

[Comment]: Figure 4: Sim-new maps should be the same size as the WOUDC and Sim-ref maps.

[Response]: We adjusted the size of Sim-new maps to be the same size as the other two in the revised manuscript.

———————————————————

[Figure]

[Figure]

**Fig. 1.** Seasonality of O3/PV (Annual mean= 1, black solid line=observed mean at all sites, grey, black dash line= observed mean at sites at latitudes north of 40N, red line = fitted by function)

[Figure]

**Fig. 2.** O3 trend in the top layers (pressure<100hPa) in the Arctic (latitude>75N)

---

## Author Comment (AC2) · 20 Jul 2016

[Comment]: The authors derive an empirical correlation between stratospheric ozone and potential vorticity from two sets of data, with PV from the 21-year WRF simulation and ozone from WOUDC radiosondes during the same period. Specifically, they fit the ratio of ozone to PV as an order-5 polynomial function of latitude and an order-2 polynomial function of pressure (height). The temporal fitting, representing the seasonal variation, is done using a sine function with adjustable amplitude and phase. The spatial fitting is applied to the annually averaged data, so that the fitted model is

given by a separable function of time and space. The applicable vertical domain is from 50-100 hPa. The parameterization is then applied to the WRF model to obtain the stratospheric ozone concentration that couples to the air quality model (CMAQ) for year-2006 simulation. The authors give two reasons why they did this study: (1) there is no stratospheric chemistry scheme available in the regional model (2) there is a wide range of measured O3/PV ratio values. The putative success of this parameterization is shown in the one-year simulation (Sim-new) where the results are compared to a reference simulation(Sim-ref) where O3/PV is simply set to be 20ppb/PVU.

[Response]: We thank the referee for the thoughtful and detailed review of our manuscript. Incorporation of the reviewer's suggestions has led to a much improved manuscript. Below we provide a point-by-point response to the reviewer's comments and summarize the changes that have been incorporated in the revised manuscript.

[Comment]: My overall impression is that the authors didn't give much thought in formulating their parameterization (as I explain below). Nor did the authors try to articulate clearly why fitting polynomial functions of higher orders will be better than using the linear correlation in the context of what we already know about stratospheric dynamics and mixing, the photochemical source/sink of ozone and the processes that lead to non-conservation of PV. In my view, if the authors reviewed the stratospheric mixing literature, they could have designed better parameterization and experiments.

[Response]: We agree with the reviewer that as with most parameterizations, the development of a generalized functional relationship between O3 and PV can be further improved. In specific, the function could be better if source and sink of O3 involved in the stratospheric dynamics had been considered in the formulation of the function. However, the design of such a function would require an extensive effort to make the function comply with or redesign the current physics and chemistry structure in the model. It should be noted that most tropospheric chemistry-transport models do not include a representation of stratospheric chemistry in part because the vertical extent (and the resolution employed in the UT/LS) is often limited and also because typical

integration time periods are inadequate to represent stratospheric chemistry impacts. The primary motivation of this study is thus to investigate the development of a practical approach to represent the potential impacts of stratosphere-troposphere exchange processes on tropospheric 3D O3 distributions. Unlike most previous tropospheric CTM-based studies that have specified O3 with a fixed scaling factor, here we have attempted to develop a more generalized functional relationship that can capture the seasonal and latitudinal variations in the O3-PV correlation, especially at the higher latitudes. To our knowledge, there are no available functional relationships that can be used for this purpose. We do acknowledge that the current parameterization has some limitation and can be further improved – see for instance the revisions incorporated in response to Referee #1 comments related to representation of UT/LS O3 in the tropics. Additionally, we feel that the suitability, evolution and performance of any parameterization should also consider practical aspects such as the model vertical grid resolution, especially in the UT/LS.

To address the reviewer's concern, we have modified the discussion in the revised manuscript as below:

(Page 7 Line 15-17) "Further improvements to the parameterization should be explored through more detailed analysis of mixing process in the UTLS, through more detailed investigation of the impact of stratospheric chemistry, and improvements in the performance of the parameterizations for conditions representative of the tropical UTLS."

[Comment]: If the authors want to make the case that this is a simple paper based on "big data" statistical approach where the data are trained to relate one variable (PV) to another (ozone) by fitting polynomial functions with no need to discuss the underlying physics and atmospheric dynamics, the authors are then compelled to work out the uncertainty range of the coefficients in Table 1. Is the 20-year data long enough to obtain stable coefficients? If the data are split into two 10-year periods, are the coefficients very different from each other and if implemented in the simulation, would the results be very different from those in the 20-year run?

[Figure]

[Response]: It is important to note that in developing the O3-PV relationship in this study, we attempted to use all possible available data. Thus we leveraged the existence of a 21-year simulated record of PV in the UTLS with corresponding O3 observations. The relationship developed can thus be considered to be "climatologically" representative rather than representing a specific time period or location. No particular year is used as a training data set. The choice of 2006 for model evaluation was simply based on the fact that this calendar year is also being used for many other assessments with the hemispheric CMAQ because of the availability of additional field campaign data sets (e.g., INTEX-B, IONS, TEXAQS). The good performance with the 2006 upper air observation in fact helps build greater confidence in applicability of the climatological O3-PV relationship. We have recently also completed simulations for the year 2010 and initial analysis suggest similar performance improvements.

Nevertheless, the reviewer raises a good point on the sensitivity of our methodology (and the derived coefficients in the proposed functions) to the length of the data record used. To further investigate the reviewer's question, we split the data into two 10-year periods, i.e., 1990-2000, and 2000-2010. As seen in Figure S8 (Figure C1), the differences between the coefficients derived using different data sets are relatively small. The functions based on data from 1990-2000 and 2000-2010 look quite similar to each other and both are similar to the one based on 21year period used in this study. Further to illustrate that the parameterization is not "trained" for a specific year, we leave out data for the year 2006, and used the remaining 20 years data to parameterize the function. The 2006-leave-out function looks very close to the full-21-years function, and the discrepancy between all coefficients is less than 20%. Therefore, the function parameterized in this study is not specific to a time or location, but rather designed to capture the average variability represented in the long-term record. This information is now included in the revised supplemental information material accompanying the revised manuscript.

To address the reviewer's concern, we clarify this point in the revised manuscript as

below:

(Page 7 Line 6-15) "It is important to note that we attempted to use all possible available data in developing the O3-PV relationship in this study. Thus we leveraged the existence of a 21-year simulated record of PV in the UTLS with corresponding O3 observations. The relationship developed can thus be considered to be "climatologically" representative rather than representing a specific time period or location. No particular year is used as a training data set. The stability of the function has been examined by leaving out the year of 2006 for parameterization, and results show that the resulting function barely changed due to this perturbation, suggesting the function parameterized in this study is not specific to a time or location, but rather designed to capture the average variability represented in the long-term record. Figure S8 presents a comparison of the O3-PV functions developed using different lengths of data records. As illustrated by the results, the inferred functions are quite similar across these different data sets, thereby providing some confidence in its robustness in representing the seasonal and latitudinal variations in O3-PV."

[Comment]: I suspect because the parameterization is confined to the region between 50 and 100hPa, for Sim-new, the free tropospheric ozone concentrations are likely biased high, particularly worse in the winter when the tropopause is the lowest. Here is my reasoning. The tropopause height changes seasonally and STE occurs at the tropopause in connection with the lowermost stratosphere. The tropopause over winter mid- and high-latitudes can be as low as 300 hPa. In other words, the parameterization completely missed the lowermost stratosphere (LMS, defined as the volume enclosed by the 380K isentropic surface/or alternatively the 100hPa isobaric surface at the top and the tropopause below) where the isentropic mixing delivers air and ozone mass across the tropopause. In my view, if one feels compelled to parameterize ozone using PV, then the crucial region where this needs to be done well is the LMS region to capture the isentropic mixing. Even we assume that the Brewer-Dobson circulation transports the correct amount of ozone to the LMS when using the authors' parameterization, and that the model does the correct mixing, one would still expect that the resulting ozone would still be biased high because of the missing midlatitude photochemical ozone loss in the LMS.

[Response]: We agree with the reviewer that indeed a parameterization such as this could be extended to lower levels, but in this initial study we did not since most regional scale models do not have sufficient vertical resolution in the LMS. In fact early sensitivity simulations, clearly showed that model results were sensitive to the vertical resolution employed (see for instance Mathur et al., 2008; available at: https://www.cmascenter.org/conference/2008/agenda.cfm). To specify O3-PV down to 300mb would also require a much finer vertical resolution. Nevertheless, we agree that the reviewer raises a good point that will be investigated in more detail in future studies and in further evolution of this parameterization. It should however be noted that photochemical loss for O3 in the portions of the LMS included in the model's vertical extent, is represented based on the detailed tropospheric chemistry mechanism used in the modeling system.

To clarify this issue, we have provided the following discussion in the revised manuscript as below:

(Page 7 Line 17-21) "A limitation of this study it that the current model setting lacks sufficient vertical resolution in the lowermost stratosphere. To minimize effects of artificial numerical diffusion associated with the current limited vertical resolution employed in the model, we limit the application of the parameterization to between 100-50mb. Future studies with a much finer vertical resolution, especially to adequately capture the seasonal variation in the tropopause height are suggested to further help evolve the O3-PV parameterization and its practical use."

[Comment]: For the reference case (Sim-ref) the specification of 20 ppb/PVU guarantees a large underestimation of stratospheric ozone as well as a large underestimation of ozone fluxes into the troposphere. It is very common in the literature that

the tropopause is defined as the surface of PV= 2 PVU or Ozone =100 or 150 ppb of ozone. These isopleths are in close proximity. This is equivalent to 50-75 ppb/PVU near 150 to 300 hPa. Thus this sim-ref specification of 20 ppb/PVU is not only way too low for 50hPa as the authors already mentioned but is just too low in general. Another way to confirm my suspicion is to check the Sim-new's coefficients at zero-order (constant term) in Table 1, which I presume is the leading term. Indeed, they are 62, 151 and 203 ppb/PVU at 95, 76 and 58 hPa, much larger than 20 ppb/PVU, even near 100hPa. Thus the major finding that Sim-new corrects the negative bias of Sim-ref but overcorrects it for the autumn and winter seasons is largely expected (Page 9).

[Response]: Yes, we agree that the 20ppb/PV unit parameterization will underestimate as it does not account for any variations in the O3-PV relationship in space and time. That precisely is the reason why we embarked on developing this parameterization.

[Comment]: It would be also helpful to see more comparisons of vertical profiles in the free troposphere alone between Sim-ref, Sim-new and WOUDC for the mean and variability of ozone. The log10(ozone) in Fig. 5 only gives a rough sense of magnitude dismatch, mainly that Sim-ref is too weak. The authors should zoom in on the free troposphere for more assessment.

[Response]: The comparisons of vertical profiles in the free troposphere between Sim-ref, Sim-new and WOUDC are given in the Figure S6 (Figure C2). The free- tropospheric ozone is significantly underestimated in Sim-ref. The low bias is reduced in the new simulation with O3-PV parameterization, particularly in mid- and high- latitude regions.

We provided the additional plots as the support information in the revised manuscript.

(Page 5 Line 27-30) "The comparison of vertical profiles in the free troposphere between Sim-ref, Sim-new and WOUDC is given in Figure S6. The free- tropospheric ozone is significantly underestimated in Sim-ref. The low bias is reduced in the simulation with the new O3-PV parameterization, particularly in mid- and high- latitude

regions."

[Comment]: It is also better to present pressure height in km or pressure in hPa instead of model layers number in Fig. 5. And more efforts are needed for caption descriptions.

[Response]: The Y scale of Figure 5 has been changed to show both "height in km and pressure in mb" as the reviewer suggested. The figure caption has also been modified, as Figure 5 (Figure C3):

[Comment]: Ultimately this paper is about surface ozone. Judged from Table 3, the surface ozone errors seem quite insensitive to the input of stratospheric ozone (Sim-ref vs Simnew). Have the authors looked into the coupling scheme between the free troposphere and the planetary boundary layer in CMAQ? Any other option for representing the boundary-layer turbulence?

[Response]: As illustrated in Table 3 and Figure 8, errors in surface O3 predictions are quite sensitive to the treatment of stratospheric ozone during spring. Compared to available measurements, the new O3-PV parameterization results in much improved model performance statistics for surface O3 during spring. The accurate representation of 3D transport mechanisms in models is critical for accurately representing the impacts of the stratosphere on lower tropospheric and boundary layer ozone – thus representation of transport by both resolved and sub-grid clouds in addition to the PBL scheme is important. The current PBL scheme in CMAQ is based on ACM2 planetary boundary layer (PBL) model (Pleim, 2007a, b). The scheme has been carefully constructed and implemented in both WRF and CMAQ to maintain consistency in the representation of mixing for both meteorological parameters as well as chemical species. It's been tested and applied in many previous studies and evaluated through comparisons with measurements of vertical profiles of various parameters. However, as suggested by the reviewer, it may be interesting to explore the use of a different PBL scheme in WRF (and CMAQ) model and will be explored in future studies.

Reference:

Pleim, J. E.: A Combined Local and Nonlocal Closure Model for the Atmospheric Boundary Layer. Part I: Model Description and Testing, J. Appl. Meteorol. Clim., 46, 1383–1395, doi:10.1175/JAM2539.1, 2007a.

Pleim, J. E.: A Combined Local and Nonlocal Closure Model for the Atmospheric Boundary Layer. Part II: Application and Evaluation in a Mesoscale Meteorological Model, J. Appl. Meteorol. Clim., 46, 1396–1409, doi:10.1175/JAM2534.1, 2007b.

[Figure]

[Figure]

**Fig. 1.** Sensitivity analysis for the PV function

[Figure]

Figure S6. Zonal mean profiles of ozone and error metrics for the different cases in free troposphere (pressure>500mb). (a) Simulated ozone profile in reference case; (b) Simulated ozone profile in new case with updated $O_3$-PV parameterization ; (c) Observed ozone profile, the annual mean of measurement time period for each WOUDC site across the northern hemisphere; (d) Normalized Mean Bias in the reference simulation; (e) Normalized Mean Bias in the new simulation with updated $O_3$-PV parameterization; (d) Difference in Normalized Mean Errors between the new simulation and reference simulation (unit: ppb, 2006 Jan-Dec; NMB-Normalized Mean Bias; NME-Normalized Mean Error; *ΔSim= Sim-new minus Sim-ref in NME)

**Fig. 2.**

[Figure]

Figure 5: Zonal mean profiles of ozone and error metrics for the different cases. (a) Simulated ozone profile in reference case; (b) Simulated ozone profile in new case with updated $O_3$-PV parameterization ; (c) Observed ozone profile, the annual mean of measurement time period for each WOUDC site across the northern hemisphere; (d) Normalized Mean Bias in the reference simulation; (e) Normalized Mean Bias in the new simulation with updated $O_3$-PV parameterization; (d) Difference in Normalized Mean Errors between the new simulation and reference simulation (unit: ppb, 2006 Jan-Dec; NMB- Normalized Mean Bias; NME-Normalized Mean Error; *ΔSim= Sim-new minus Sim-ref in NME)

**Fig. 3.**